# Effects of Walnut and Pumpkin on Selective Neurophenotypes of Autism Spectrum Disorders: A Case Study

**DOI:** 10.3390/nu15214564

**Published:** 2023-10-27

**Authors:** Afaf El-Ansary, Laila Al-Ayadhi

**Affiliations:** 1Autism Center, Lotus Holistic Alternative Medical Center, Abu Dhabi P.O. Box 110281, United Arab Emirates; 2Autism Research and Treatment Center, P.O. Box 2925, Riyadh 11461, Saudi Arabia; lyayadhi@ksu.edu.sa; 3Department of Physiology, Faculty of Medicine, King Saud University, P.O. Box 2925, Riyadh 11461, Saudi Arabia

**Keywords:** autism spectrum disorder, neurophenotypes, pumpkin, walnuts, glutamate excitotoxicity, nutritional intervention

## Abstract

Special diets or nutritional supplements are regularly given to treat children with autism spectrum disorder (ASD). The increased consumption of particular foods has been demonstrated in numerous trials to lessen autism-related symptoms and comorbidities. A case study on a boy with moderate autism who significantly improved after three years of following a healthy diet consisting of pumpkin and walnuts was examined in this review in connection to a few different neurophenotypes of ASD. We are able to suggest that a diet high in pumpkin and walnuts was useful in improving the clinical presentation of the ASD case evaluated by reducing oxidative stress, neuroinflammation, glutamate excitotoxicity, mitochondrial dysfunction, and altered gut microbiota, all of which are etiological variables. Using illustrated figures, a full description of the ways by which a diet high in pumpkin and nuts could assist the included case is offered.

## 1. Introduction

Autism spectrum disorder (ASD) is a collection of neurodevelopmental disorders that have deficiencies in two behavioral domains: (i) social interaction and communication difficulties and (ii) restricted interests and repetitive and stereotyped behaviors (DSM-5) [1,2,3]. Seizures, attention-deficit/hyperactivity disorder (ADHD), and other cognitive deficits are also possible co-occurring illnesses with ASD [3,4]. ASD affects around 1% of the human population [5], with males affected four times more frequently than females [6]. A well-balanced diet is rich in nutrients. This can be difficult for autistic people because many of them have digestive and eating challenges. Children with autism usually have poor nutrition, owing in part to their food avoidances. Autistic children are more likely to have low calcium and protein levels, which can impair brain development, bone growth, and muscle strength. These impairments could be linked to challenges with cognition, balance, physical strength, and other areas of physical development. Fiber, folic acid, calcium, iron, zinc, and vitamins A, C, D, E, K, B6, and B12 were found to be the most commonly deficient nutrients in children with autism. Feeding concerns can be a big issue for autistic children, with catastrophic consequences if the child suffers from nutritional deficits [6]. The concomitant lower concentrations of vitamins B6, B9, and B12 may cause methylation deficiency in autistic children [7,8,9,10,11].

Individuals with ASD frequently display aberrant amounts of amino acids, which indicate disrupted amino acid-dependent activities. Food limits, restrictive diets high in protein, and impaired digestion are the root causes of these disorders [12]. There are reports in the literature of abnormal tryptophan levels [13,14], as well as altered levels and expression of serotonin, glutamic acid, gamma-aminobutyric acid (GABA), and homocysteine [15,16]. According to research, people with ASD may be deficient in tryptophan. Tryptophan deficiency was determined by looking at Trp levels or its primary metabolite kynurenine in urine [17].

An earlier analysis of Children’s Autism Metabolome Project (CAMP) data revealed an inverse relationship between a group of plasma metabolites in autistic children and plasma branched-chain amino acids (BCAAs) [18]. Amino acid metabotypes (AADMs), which were present in 17% of the ASD individuals, were identified as a result of imbalances in glycine, glutamine, and ornithine concentrations in relation to BCAA concentrations [19].

Dietary bioactive peptides are an important nutritional component. A dietary bioactive peptide with the potential to exert therapeutic effects on the brain requires a long time to reach its target region. Before ingestion, the peptide might be generated through fermentation. The meal protein or peptide must pass through the entire gastrointestinal tract after ingestion, where it is extremely susceptible to breakdown by acidic stomach conditions and proteolysis by digestive enzymes. Peptide transport 1 (PepT1) has been identified as the major mechanism for di/tri peptide transport across the intestinal epithelial membrane [20]. Peptide transport may be aided by other mechanisms such as paracellular, transcytosis, and transcellular diffusion. The absorption pathway can be influenced by the size, hydrophobicity, and net charge of peptides [21]. Peptides have relatively short half-lives in the blood and may only remain functionally intact for short periods of time (minutes to hours) [22]. It is unknown how peptides pass the blood–brain barrier (BBB); however, it could be by receptor-mediated transcytosis or through specific transporters [23].

According to several research findings, dietary hydrolysates have higher antioxidant activity than pure peptides, indicating a synergistic impact between free amino acids and peptides [24]. Furthermore, short-chain peptides appear to have greater antioxidative capability than higher molecular weight equivalents [25].

For thousands of years, people have utilized walnuts (*Juglans sregia*) as a nutritious food. Numerous studies have demonstrated the potent antioxidant and anti-inflammatory properties of walnuts. Unsaturated fatty acids, peptides, proteins, and phenolic substances that have anti-inflammatory characteristics are abundant in walnut kernels [25]. In rats and people, walnut consumption has been shown to enhance memory function and other cognitive processes [26].

Pumpkins belong to the family of cucurbitaceas, known for their high content in bioactive compounds, with benefits for human health [27]. Interestingly, because pumpkins are high in nutraceuticals and functional ingredients, their consumption and processing as pharma food should be encouraged due to their antihyperlipidemic, antiviral, anti-inflammatory, antihyperglycemic, immunomodulatory, antihypertensive, antimicrobial, and antioxidant potential. Pumpkin has a domain in the form of powders, extracts, and pumpkin-infused cooking products. Pumpkin has been used to produce an assortment of healthy, nutritious, and functional food products, including juice, soup, oatmeal, chips, biscuits, bread, cake, bars, and noodles. In recent years, some new and novel technologies have been used to prepare and preserve pumpkin for increased shelf life and nutrient bioaccessibility [28].

Carotenoids such as violaxanthin, astaxanthin, antheraxanthin, zeaxanthin, lutein, lycopene, and β-carotene, as the major nutraceuticals ingredients in pumpkin, are naturally occurring antioxidants that individuals can eat [29].

Based on our understanding of the etiology of ASD and the remarkable improvement of an autistic child after three years of consuming a pumpkin/walnut-rich diet, gluten- and casein-restricted diet, and GABA supplementation over a three-year duration, it was exciting to investigate the neuroprotective and therapeutic effects of walnuts and pumpkin on specific signaling pathways in ASD. Special reference will be given to pumpkin/walnut-rich diet effects because the effect of GABA supplementation and gluten and casein restriction has already been explored in ASD. The inclusion of a case study adds another dimension to our review as it provides actual results in terms of the effects of using pumpkin- and walnut-rich diets as a nutritional intervention on a young boy with moderate autism. The information for the case study was gathered through face-to-face contact with Child-X’s mother, who was willing to share her son’s intervention and cure story. This review aims to provide an informative and explorative study into the effects of a pumpkin/walnut-rich diet, concomitant with a gluten/casein-restricted diet and GABA supplementation, on a child with autism and to increase the awareness of dietary supplements. The follow-up of Child-X’s improvement was conducted by the measurement of his Childhood Autism Rating Scale (CARS) score. The CARS score was examined as a gauge of autism severity. A scale of 1 to 4 is used to rate a child on a scale of 15 traits or behaviors, including verbal communication; listening response; fear or nervousness; imitation; body use; object use; ability to relate to others; emotional response; nonverbal communication; activity level; level and dependability of intellectual response; adaptation to changes; visual response; taste, smell, and touch responses; and general impressions. A total score of at least 30 strongly suggests the presence of autism. Children with scores between 30 and 36 have mild-to-moderate autism, whereas those with scores between 37 and 60 have severe autism [30].

## 2. Case Study

The participant in the case study, Child-X, is now a 13-year-old boy known to the author who scored 22 on the CARS test (for autism). Before the three-year dietary intervention described above, his CARS score was 36 and he was diagnosed with moderate autism. The author elected to write up a case study of Child-X to provide a first-hand experience of the effects of the dietary intervention. Child-X’s mother was also included in the collection of information for this case study.

### 2.1. Detailed History of the Case Study

Child-X was the second child in his family. Child-X’s sleep had been brief and infrequent since birth, and when he began to speak, it was evident that his linguistic development was severely impaired. His speech was a mumble.Child-X enjoyed playing alone in a corner, arranging his cars from smallest to largest or arranging shoes. He did not respond to his name, so if called, he would not come, and if hugged or caressed, particularly his hair, he would not respond (Child-X’s mother stated).When Child-X was about a year and a half old, he began to pronounce “mama” and “papa,” but he did not mean either and he had no reaction to pain, even throughout the teething stage.Child-X was often frightened and could injure himself by crashing his head against the ground or a wall till he was harmed. He threw objects on the ground, and if they shattered, he went into a two- to three-hour-long panic attack. He slept on the ground, or even in the street, and he did not hurt anyone, but if anyone approached or touched him, he reacted violently. Child-X underwent a hearing test, which came out normal.Child-X went to a speech–language therapist when he was two years and two months old and, after careful examinations, she diagnosed him with moderate autism, recommending that he see a neurologist and go through speech, skill development, and behavior management sessions. The neurologist confirmed that he had moderate-grade autism and that he would never be normal.

### 2.2. Dietary Intervention

Child-X was on a gluten- and casein-restricted diet and his mother changed to soy and almond products. To activate brain cells, the doctor suggested including pumpkin and walnuts in the diet.The therapy journey then began with Omega 3, GABA 500 mg/day, and a sedative, which was gradually tapered off because his parents did not want him to sleep for long periods of time. GABA was administered for only one year of the three years of the pumpkin/walnut-rich diet regimen. For over three years, the normal diet of child-X included blended mango juice with pumpkin, cooked pumpkin or soy products mixed with lentil and carrot, and daily servings of cake or pastries topped with chopped walnuts. Pumpkin was also used to make muffins, custards, and pancakes.Along with the food intervention, Child-X received speech, behavioral, and educational therapy twice a week for three years. A speech therapist measured the improvements and recorded them. Child-X’s mother recorded a remarkable improvement in speech going forward.

### 2.3. Language Development and CARS Score

Then, at the age of four and a half, Child-X began to call his mother and say simple words without making usable phrases, and he began to speak intentionally and intelligibly at a rate of 30%. Now, he is speaking normally and, through behavioral and social interaction training, his CARS score has continued improving to reach 17. Table 1 displays the improvement in his CARS score during the course of the dietary intervention (from 2 years and 2 months to 6 years) and up until age 13 (Child X continued to prefer eating pumpkin and walnuts frequently).

Child-X is now diagnosed as “out of autism” and he is being integrated into a regular school and achieving satisfactory learning outcomes in grade six. His mother was very keen to share her son’s management story with parents of individuals with ASD.

## 3. Etiological Mechanisms of ASD

The term “treatment” is highlighted as many people argue that autism is not a treatable disorder. Considering the great progress in understanding the etiological mechanisms of ASD, it is very interesting to explain the scientific evidence of how a pumpkin/walnut-rich diet was effective in treating Child-X.

### 3.1. Glutamate Excitotoxicity in ASD

The presence of shared symptoms among ASD cases, despite their great degree of variation, points to shared impairments in a few neurological pathways. The development of the peripheral and central nervous systems depends heavily on neurotransmitters. So, it is possible that neurotransmitter dysfunctions have a role in the pathogenesis of ASD. Since it is directly involved in brain development and synaptogenesis [31,32,33], memory, behavior, and motor activity control [34,35,36], as well as gastrointestinal processes [37,38], glutamate (Glu) is regarded as a good biomarker among neurotransmitters [31,32,33].

Glutamate, the major excitatory neurotransmitter in the mammalian central nervous system, has multifarious functions in the brain such as the detoxification of ammonia, as an important building block in the synthesis of proteins and peptides including glutathione, as a precursor for the synthesis of an inhibitory neurotransmitter GABA, the synthesis of glutamine (Glx) by glutamine synthetase, and cleaving glutamate from glutamine by glutaminase [39].

Important evidence about essential alterations in Glu content in both pediatric and adult individuals with ASD is provided by studies on postmortem or bodily fluid samples [37,38]. Excitotoxicity has already been generally recorded as an etiological mechanism linked to ASD phenotypes. Additionally, the higher rate of seizures in autistic individuals compared to healthy subjects supports the idea that excitatory and/or inhibitory network activity is dysfunctional [40,41,42].

In the mammalian brain, Glu is the primary excitatory and GABA is the main inhibitory transmitter, respectively. According to estimates, the intracellular compartment of the CNS contains about 10 mmol/L of Glu, which is substantially greater than the extracellular fluid’s concentration of 0.5–2 mmol/L [43,44], the cerebrospinal fluid’s concentration of 10 mmol/L, or the plasma’s concentration of 150 mmol/L [45]. Brain Glu content is closely regulated by a variety of processes, including the blood–brain barrier’s endothelial cells and the Glu/Glx/GABA cycle between neurons and astrocytes, as disruptions in the Glu/GABA system can have harmful effects and are related to the severity of ASD. Moreover, microglia have an important contribution to glutamate signaling in ASD [46].

The severity of ASD has been linked to higher plasma levels of Glu [47,48,49]. In affected individuals, it has been described as increased Glu/Glx and Glu/GABA ratios [50]. ASD individuals’ plasma Glu levels as well as specific brain areas have both been found to be altered [32]. Although saturable and stereoselective facilitative transporters on the luminal membranes prevent significant amounts of Glu from entering the brain [44], blood Glu levels could theoretically affect brain concentration; furthermore, Glu dysregulation is related to blood–brain barrier (BBB) permeability and should be considered as a possible therapeutic target [51].

Many processes contribute to GE and neuronal death. Under physiological conditions, Ca^2+^ enters neurons via the *N*-methyl-D-aspartate (NMDA) receptor ion channels. High Ca^2+^ concentrations cause neuronal cell death in a variety of ways, the most damaging being Ca^2+^’s effect on the mitochondrial membrane potential (MMP). Ca^2+^ overload can also trigger the generation of free radicals and the upregulation of harmful transcription factors. Ca^2+^-dependent enzymes are also activated, including phospholipase A2, cyclooxygenase-2, lipoxygenases, protein kinase C, calpain, xanthine oxidase, proteases, endonucleases, phospholipases, and lipases [52].

In multiple ASD rodent models, there is strong evidence that blocking mGluRs can reverse autistic traits. Since glutamate receptors play a significant role in autism, various treatment approaches that target these receptors are being explored to lessen ASD symptoms. To correct the excitation and inhibition balance in the cortical regions of the brain, a number of mGluR-targeting medications are employed [32,53]. In rat models of ASD, Glu5 antagonists enhance social and stereotypical behaviors [54]. In a BTBR mouse model of autism, the mGluR5 antagonist MPEP reduced repetitive self-grooming, whereas an mGluR5 antagonist GRN-259 reduced repetitive behaviors in three cohorts of BTBR mice [55].

The Xc system is used by microglia to participate in glutamate signaling, and the Xc transporter is a chloride-dependent antiporter that has the ability to transport glutamate outside of the cell. Produced by microglia, ROS cause glutathione (GSH) depletion, trigger the TLR4 signaling pathway, enhance Xc expression, and cause glutamate efflux. Microglia release ROS, IL-1, and TNF, which inhibit excitatory amino acid transporter (EAAT) activity and raise extracellular glutamate levels. As a result of decreased glutamate uptake, excitotoxins such as glutamate, D-serine, and ATP are released, changes in the glial transmitter release from astrocytes, and reactive microglia actively interfere with neurotransmission [56]. It is unknown if microglia actively contribute to the development of the inhibitory circuit. However, GABA-receptive microglia selectively remove inhibitory synapses throughout development.

### 3.2. Oxidative Stress and ASD

Children with ASD have been found to have elevated oxidative stress markers, including decreased antioxidant enzyme activity, increased lipid peroxidation, and an accumulation of advanced glycation products in peripheral blood, as well as mitochondrial dysfunction [57,58,59]. The autistic brain shows differential expression of apoptosis-related proteins localized on the mitochondrial membrane, indicating an increased apoptosis of brain cells throughout development [60]. This is consistent with the abnormal brain development that has been shown in ASD-affected children. Mitochondrial dysfunction has been identified in ASD individuals [61,62,63] and it is thought to be one of the reasons for ASD pathology. In fact, oxidative stress is enhanced in the brains of ASD individuals due to anomalies in the mitochondrial electron transport system [64,65].

### 3.3. Neuroinflammation in ASD

Inflammation is thought to exacerbate pathological abnormalities that underpin ASD, and prior investigations have found higher inflammation in children with ASD [66,67]. For instance, it has been reported that the peripheral serum and plasma of children with ASD contain inflammatory cytokines like transforming growth factor-1 (TGF-1), hepatocyte growth factor (HGF), epidermal growth factor (EGF), platelet-derived growth factor-BB (PDGF-BB), and tumor necrosis factor (TNF). Additionally, it has been noted that children with ASD have greater levels of IL-1, IL-4, IL-6, IL-8, and IL-13 in their peripheral serum and plasma [68,69]. In addition, children with ASD have higher levels of inflammatory chemicals in their postmortem brain and cerebrospinal fluid (CSF), including IL-1, IL-6, TNF, and monocyte chemoattractant protein-1 (MCP-1). Additionally, it has been observed that levels of IL-1 and IL-4 in blood samples from newborns are related to the severity of ASD. These findings show that the pathophysiology of children with ASD is connected to a systemic inflammatory response [70].

### 3.4. Mitochondrial Dysfunction as a Central Etiological Mechanism in ASD

The pathophysiological causes underlying ASD are complex. They are mostly murky. However, the pathophysiology of ASD and the mitochondrial hypothesis are becoming clearer. ASD individuals’ damaged neurons often exhibit mitochondrial malfunction, suggesting a crucial role for mitochondrial dysfunction in the pathophysiology of ASD. By restoring normal mitochondrial processes, ASD can be treated with a variety of changes in the brain [71].

Mitochondria can be found in all synaptic compartments, including the presynaptic terminal, the dendritic spine’s base, the astrocytic body and processes, and microglial cells (when regarded as a synapse). In the presynaptic terminal, mitochondria produce energy (ATP) for synaptic vesicle production, neurotransmitter release into the synaptic cleft, and mediator reuptake. In dendrites, mitochondria are found at the base of the most functionally loaded dendritic spines, where they provide energy for the main stages of synaptic plasticity: phosphorylation, receptor externalization and synaptic recruitment, and changes in dendritic spine structure. Ca^2+^ homeostasis, as well as its global and local gradients, are determined by the quality of the mitochondrial state at the base of the dendritic spine, including membrane potential [72,73,74]. In astrocytic processes, mitochondria help glutamate transporter-1 (GLT-1) and excitatory amino acid transporter (EAAT) 1 and 2 operate, among other things [75,76,77].

The electron transporters ubiquinone and cytochrome c are part of the mammalian electron transport chain (ETC), which also consists of complexes I through IV. According to Guo et al. [78], the ETC is the location where proton gradient generation and electron flow are connected across the inner membrane, and complex V (ATP synthase) uses the energy that is accumulated throughout this process to make ATP. A number of neurodegenerative disorders are linked to eventual ETC complex deficits. Multiple studies have reported that variations in mitochondrial ETC complex levels may play a role in the etiology of autism. As a result, oxidative stress and impaired energy metabolism will occur in autism [79,80,81].

### 3.5. Autophagy and ASD

Cell viability depends on autophagy, a tightly controlled and conserved biological process. It maintains cellular homeostasis under stressful environmental circumstances by encouraging the recycling of organelles and durable proteins [82]. To regulate autophagy, neurons have evolved incredibly specialized mechanisms [83]. Early synaptic pruning, a developmental process in which over 70% of postnatal net spines are destroyed to ensure the relevant creation of suitable neuronal connections, depends on neuronal autophagy [84]. Neuronal autophagy dysfunction has been associated with neurodegenerative diseases like Parkinson’s and Alzheimer’s disease and neurodevelopmental conditions like autism spectrum disorder (ASD) [85,86]. It is intriguing to draw attention to the role that functionally relevant polymorphisms in genes related to autophagy play in the susceptibility to autoimmune and inflammatory conditions that are known to be linked to ASD [87,88]. Beyond its crucial function at the CNS level, autophagy appears to have another purpose that is pertinent to NDDs. Autophagy does, in fact, have a demonstrable function in intestinal homeostasis, influencing both cell metabolism and proliferative and regenerative abilities [89]. These recent findings are especially intriguing in light of the fact that (i) the gut microbiota affect brain function via the neuroendocrine, neuroimmune, and autonomic nervous systems as well as through the production of microbiotic toxins and (ii) both inflammation and the gut microbiota may play a significant role in the pathophysiology of NDDs [90,91,92].

### 3.6. Altered Gut Microbiota in Autism

The “leaky gut” concept proposes that deficiencies in intestinal epithelial barrier permeability cause improper signaling by luminal components such as bacteria, environmental toxins, and even food macromolecules. Increased intestinal permeability permits bacteria-secreted compounds to pass the barrier and enter the blood, where they can impact the brain by boosting cytokine release and initiating an immunological response [93]. Inflammatory cytokines generated by immunological activity influence the CNS and disrupt normal brain development early in life, potentially leading to ASD through the gut–brain axis [94].

Maternal prenatal drug use, maternal health variables, and prenatal infection have all been linked to the development of ASD [95]. ASD is also linked to an increased prevalence of gastrointestinal (GI) problems [96,97]. Chronic constipation, diarrhea, abdominal pain, and potential indicators of GI inflammation such as vomiting and bloody feces are examples of GI difficulties [98]. Constipation has also been linked to distinct bacterial patterns in autistic and neurotypical participants, with constipated autistic people having high numbers of bacterial taxa from the Escherichia/Shigella and Clostridium cluster XVIII. It was also found that the relative prevalence of the fungus genus Candida was more than twofold in autistic subjects compared to neurotypical subjects [99]. Food-based administration to Lactobacillus reuteri or Bacteroides fragilis improved ASD-like social impairments in mice in animal studies [99,100]. In humans with ASD, there is evidence of dysbiosis of the gut microbial population and notably changed levels of Bifidobacterium, Lactobacillus, and Clostridium species [101,102]. Furthermore, fecal microbiota transplantation in children with ASD improved GI and ASD symptoms [103,104]. Probiotic use has also been shown to reduce core ASD symptoms [105].

## 4. Walnut/Pumpkin Therapeutic and Neuroprotective Properties

### 4.1. Antioxidant and Anti-Inflammatory Effects of a Walnut/Pumpkin-Rich Diet

Numerous lines of evidence suggest that walnuts (*Juglans regia* L.) may have a good impact in treating brain disorders due to the additive or synergistic effects of their antioxidant and anti-inflammatory components. Walnuts are rich in antioxidants, including flavonoids, phenolic acid, melatonin, folate, vitamin E, selenium, juglone, and proanthocyanidins. Walnuts are high in omega-3 fatty acids and alpha-linolenic acids (ALAs), which are a precursor to DHA and EPA [106,107]. However, ALA is poorly converted to EPA, despite the fact that ALA has been demonstrated to improve brain function [108]. Walnuts are also high in fiber, vitamins, minerals, and bioactive substances that have been shown to improve brain health [106]. In humans, short-term walnut eating has been shown to raise peripheral EPA levels, and various experimental trials have suggested possible cognitive benefits [109].

Although the majority of nuts contain monounsaturated fats, only walnuts (13 g of the 18 g total fat per ounce of walnuts) primarily contain polyunsaturated fat, with 2.5 g of ALA. Eicosapentaenoic acid (EPA) and docosahexaenoic acid (DHA), which are known to have anti-inflammatory properties, are precursors to ALA. According to research, ALA reduces inflammation by downregulating iNOS, which prevents the generation of NO, COX-2, and pro-inflammatory cytokines such IL-1, IL-6, and TNF-α [110,111,112,113].

Walnut consumption (1–2 oz daily) may enhance cognitive function and a long-term dietary supplementation of walnuts can (a) significantly enhance memory function, learning abilities, motor coordination, and anxiety-related behavior and (b) attenuate oxidative stress by improving the balance between free radicals and antioxidants and associated neuronal cell death. Studies point to the potential benefits of early and ongoing nutritional interventions using walnuts for preserving cognitive function and suggest the possibility of using walnuts as a treatment intervention for ASD [114].

In LPS-induced RAW264.7 cells, epicatechin (EC), another major catechin in walnuts, was found to downregulate the proinflammatory cytokines IL-1, IL-6, and TNF- α [115]. Quercetin, a member of the flavonol family of chemicals prevalent in walnuts has been shown to lower iNOS expression in LPS-activated BV-2 microglia and inhibit NF-κB activation, showing its potential to alleviate inflammatory conditions of the central nervous system [116]. Furthermore, quercetin decreased the synthesis of proinflammatory cytokines such as TNF-, IL-1, and IL-6, as well as the development of cyclooxygenase and lipoxygenase in mast cells [117]. Quercetin has also been shown to decrease TNF-α, IL-6, and IL-1 production in LPS-activated human mononuclear U937 cells [118].

It is well-accepted that peptides that come from natural sources are powerful antioxidants. However, particularly in peptide-mediated autophagy, their precise mechanisms are still not completely understood. The mammalian target of rapamycin (mTOR) participates in a variety of bodily signaling pathways to control cell division, autophagy, and apoptosis. In mouse models, Yui et al. [119] recorded behavioral impairments associated with ASD and their recovery brought on by the mTOR inhibitor rapamycin. Multiple mTOR signaling-related genes have increased transcription in Tsc2+/− mice, indicating a critical function for dysregulated mTOR signaling in the ASD model. Crucially, one study has suggested a link between postmortem dendritic synaptic surplus in ASD and hyperactivity of the mTOR pathway [120]. mTOR is a key regulator of synaptic protein synthesis [121], and aberrations in mTOR signaling have been linked to synaptic and neuroanatomical abnormalities that are associated with ASD [122]. The mTOR inhibitor may be helpful for the pharmacological treatment of ASD.

In a study conducted by Zhao et al. [123], the authors reported that three walnut peptides, TWLPLPR, YVLLPSPK, and KVPPLLY, demonstrated an inhibitory effect on mTOR and antioxidative effects through the prevention of reactive oxygen species (ROS) production, elevation of glutathione peroxidase (GSH-Px), elevation of adenosine 5′-triphosphate (ATP) levels, and amelioration of apoptosis in Aβ25–35-induced PC12 cells.

In order to protect neuronal and microglial cells from oxidative and inflammatory stress, walnut hydrolysates have been widely researched [124,125,126,127]. Numerous sequences of walnut peptides have been found to suppress the inflammatory pathway (NF-κB/p38 MAPK) and activate the protective pathway (Nrf2/HO-1) to cause the downregulation of cytokines (TNF-α, IL-1b, and IL-6) in neurons and microglial cells.

Shorter walnut peptides (LPF, GVYY, and APTLW) that are more resistant to digestion have reduced the leakage of inflammatory mediators such as nitric oxide (NO) and prostaglandins (PGE2), followed by a decrease in the expression of its corresponding enzymes, inducible nitric acid synthetase (iNOS) and cyclo-oxygenase-2) COX-2, in an LPS-stimulated microglia cell line [125,127]. Hydrophobic amino acids in walnuts like Leu (L), Pro (P), Val (V), and Ala (A) and aromatic amino acids like Phe (F), Tyr (Y), and Trp (W) have been linked to anti-inflammatory effects [127]. Xanthine oxidase inhibition and antioxidant properties of walnut-protein-derived peptides are enhanced by tryptophan residue in vitro [128]. The suggested antioxidant and anti-inflammatory properties of walnut peptides are illustrated in Figure 1.

Pumpkin (Cucurbita moschata) is one of the well-known edible vegetables and is consumed directly or processed into NF-B, MAPK products [129]. Due to its nutritional components, which have antioxidant, antifatigue, and anti-inflammatory effects on stroke or traumatic brain damage as well as in neurodegenerative illnesses, pumpkin has drawn a lot of attention. Amino acids, which are precursors to neurotransmitters, are among the most prevalent nutritional deficits identified in people with mental disorders [130]. L-tryptophan is an important amino acid found in pumpkin. The latter, as well as its intermediate metabolite 5-hydroxytryptophan (5-HTP), participate in the synthesis of the neurotransmitter serotonin and are thus recommended for ASD as a disorder related to abnormal serotonin metabolism [131]. Recent research suggests that interactions between serotonin and other systems, such as oxytocin, may be especially crucial for social behavior in ASD. These findings suggest the serotonin system as a promising therapy target [132].

It should be noted that, in general, there is an increase in serotonin in ASD, which means that the reference to the tryptophan being provided by pumpkins may be a little misleading. At this point, it would be helpful to clarify that the problem with the tryptophan-melatonin pathway in ASD appears to be the decreased ability to use serotonin to initiate the melatonergic pathway; consequently, the increase in serotonin in ASD is usually linked to a decrease in melatonin in various body cells, such as pinealocytes, platelets, and intestinal epithelial cells [133]. Given that the melatonergic pathway is found in the mitochondria of every body cell that has been studied to date, this is most likely connected to the changes in mitochondrial activity in ASD [134]. Because of differential control by microRNAs (particularly miR-451), there appears to be a decrease in 14-3-3 isoforms, which accounts for the reduced availability of the melatonergic pathway in ASD. Since pumpkin and walnuts both strongly regulate a number of miRNAs [135,136], some of walnuts’ effectiveness may be related to how the tryptophan–melatonin pathway is regulated.

Additionally, because of the flavonoids contained in the pulp, pumpkin extract has displayed protective activity against cellular neuroinflammation produced by mycotoxins [137]. Pumpkin extract (PE) consumed either orally or topically has reduced contact dermatitis-associated depression phenotypes by downregulating mRNA levels of pro-inflammatory cytokines such as TNF-α, IL-6, COX-2, and iNOS while upregulating antioxidants [138].

Furthermore, astaxanthin as a major carotenoid in pumpkin has been shown to reduce microglial activation and the release of pro-inflammatory cytokines. This chemical is also known to maintain neuronal integrity [139,140,141]. Astaxanthin was demonstrated to alleviate behavioral abnormalities and oxidative stress in a prenatal valproic acid (VPA)-induced mice model of ASD [142]. The confirmation of these findings in autistic individuals means that they may be advised to consume astaxanthin-rich foods to alleviate ASD-like symptoms. The anti-inflammatory effects of pumpkin are shown in Figure 2.

### 4.2. Walnut/Pumpkin and the Treatment of Glutamate Excitotoxicity

Certain autistic features have been linked to neuronal apoptosis that is mediated by glutamate excitotoxicity, and the reversal of this toxicity could be an excellent target to treat ASD clinical presentation [42]. PC12 cells have been used extensively in neurological research because of their resemblance to neurons in terms of shape and physiological function [143,144]. Glutamate, a significant neurotransmitter, is essential for the transmission of signals in neurons and glia. Extracellular glutamate can damage neuronal cells, though, since it promotes the buildup of reactive oxygen species (ROS) and prevents glutathione formation. Additionally, it has been suggested that mitochondrial activity and calcium homeostasis play a role in glutamate excitotoxicity [145].

Wang et al. [146] investigated the neuroprotective effects of various walnuts peptides, including WSREEQ, WSREEQE, WSREEQEREE, ADIYTE, ADIYTEEAG, and ADIYTEEAGR, on glutamate-induced cytotoxicity in PC12 cells. While overproduction of ROS and Ca^2+^ influx was observed after treatment with glutamate alone, in PC12 cells, as reported by other researchers’ treatment with walnuts peptides, exert anti-excitotoxicity effects through multiple mechanisms. Among these mechanisms are the following. (1) The reduction of Ca^2+^ influx and the concomitant generation of ROS. Therefore, the reduction of oxidative stress in glutamate-induced PC12 cells could partially explain how walnut peptides (WSREEQEREE and ADIYTEEAGR) limit Ca^2+^ influx and even NMDA receptor activation as two signaling processes in ASD. (2) In PC12 cells, glutamate insult clearly enhanced the amount of lipid peroxides while decreasing antioxidant enzyme activities (SOD and GSH-Px). Simultaneously, the clear opposite effects of WSREEQEREE and ADIYTEEAGR incubation were observed, suggesting that walnut peptides might be regarded as antioxidants and protect PC12 cells from oxidative damage. (3) Furthermore, it is commonly acknowledged that the Keap1-Nrf2 pathway is essential for controlling the cellular antioxidant system. Nrf2 promotes the transcription of genes encoding antioxidant enzymes by acting as a transcription factor that dissociates from Keap1 and enters the nucleus [147]. (4) Walnut peptides (WSREEQEREE and ADIYTEEAGR) might exert their neuroprotective effects on glutamate-induced apoptosis via amelioration of impaired mitochondrial membrane potential (MMT), inhibition of the mitochondrial apoptotic pathway, correction of mitochondrial dysfunction, and induction of mitophagy etiological mechanisms in ASD. The anti-excitotoxic effects of walnut peptides are illustrated in Figure 3.

Carotenoids are known to have positive effects on the CNS, and the most active carotenoids, astaxanthin, and lycopene, are still safe at higher concentrations. Astaxanthin (AXT), an orange-red carotenoid, is one of the major ingredients of pumpkins [148]. AXT is a fat-soluble compound that effectively crosses the blood–brain barrier. It also preserves mitochondrial function, has more antioxidant action than vitamin E, and is more potent than other carotenoids [149,150,151,152]. AXT protects neurons from excitotoxic stress by decreasing intracellular calcium increase and ROS in PC-12 and SH-SY5Y cells [152,153,154].

Kandy et al. [155] showed that the long-term administration of AXT to cortical neurons reduces the primary and irreversible secondary sustained [Ca^2+^]i response; controls the permeability of glutamate NMDA, AMPA, and KA receptors; decreases the protein expression of NMDA and AMPA receptors; inhibits the accumulation of calcium in the mitochondria; prevents the abnormal production of ROS; and promotes neuronal survival.

There are additional mechanisms by which AXT can protect cells against glutamate cytotoxicity. AXT reduced glutamate release induced by 4-aminopyridine (4-AP) in a rat cerebral cortex in a dose-dependent manner [156].

By boosting the mRNA levels of parvalbumin and calbindin D28k, two buffering proteins that reduce the overall amount of free cytosolic Ca^2+^ by binding cytoplasmatic calcium ions, AXT can also alter calcium homeostasis [157].

Moreover, lycopene as a pumpkin ingredient decreases glutamate release from rat cortical synaptosomes via reducing presynaptic Ca^2+^ entry and protein kinase C activity [158]. Anti-excitotoxic effects of pumpkin are shown in Figure 4.

### 4.3. Walnut/Pumpkin and Correction of Mitochondrial Dysfunction in ASD

Walnuts are one of the most significant sources of polyphenols among regular meals and drinks; therefore, their impact on human health demands consideration. It is very interesting that walnuts contain the polyphenol pedunculagin, also known as ellagitannin. Ellagitannins are hydrolyzed into ellagic acid after ingestion, which the gut microbiota then transform into urolithin A and additional derivatives such urolithins B, C, and D. Urolithin A activates Nrf2, which helps to regulate mitochondrial biogenesis and mitophagy as an important process for the removal of damaged mitochondria [159,160,161,162]. There is evidence that urolithin A stimulates AMPK pathway activation, which prevents apoptosis in cells with damaged mitochondria and guides metabolic processes towards mitophagy [163,164,165]. Thus, the stimulation of mitophagy, a process by which damaged mitochondria are recycled to permit renewal with healthy mitochondria, is the major signaling effect of urolithin A [166,167]. Figure 5 demonstrates the effect of walnuts in ameliorating mitochondrial dysfunction.

ASD is characterized by the breakdown of the BBB. Because of the rupture of the BBB, cells, viruses, and neurotoxic debris can enter the brain [168]. According to a new study, Lycopene, a key carotenoid found in pumpkins, is excellent at promoting the expression of ETC complex I genes in the BBB of neuronal cells, which means it may be used to treat mitochondrial failure, a neurophenotype associated with ASD [71,80,169]. Fregamione and colleagues [30] declared that pumpkin extract can assist the mitochondria in functioning more efficiently and actively by significantly increasing the expression of the complex I gene and downregulating a number of complex genes that contribute to the toxicity of the mitochondria and harm the entire cell. Figure 6 illustrates the suggested mechanism through which pumpkin can treat mitochondrial dysfunction.

### 4.4. Walnut/Pumpkin and Healthy Gut Microbiota

Consuming walnuts enhanced the probiotic-type bacteria, such as Lactobacillus, Ruminococcaceae, and Roseburia, while decreasing the bacteria associated with neuroinflammation, such as Bacteroides and Anaerotruncus [170]. Additionally, several bioactive walnut components have been shown to ameliorate gut microbial dysbiosis. A study showed that walnut oil exerted anti-inflammatory effects by decreasing the expression of TNF-α in the duodenal mucosa of mice, and the relative abundance in gut microbiota shifted from more pathogenic bacteria, such as Helicobacter and Clostridiales, towards the probiotic Lactobacillus [171]. Additionally, it was discovered that the polyphenol extract from walnut meal dramatically reduced the levels of serums LPS, TNF-, and IL-6 and prevented alterations in the intestinal flora in feces, primarily Firmicutes, Bacteroidetes, and Proteobacteria [172,173]. Additionally, consuming walnut meal dietary fiber (WMDF) significantly reduced the dysbiosis of the gut microbiota brought on by a high-fructose diet. This resulted in a rise in the relative abundance of Firmicutes, Actinobacteria, Proteobacteria, Deferribacteres, Tenericutes, and Patescibacteria and a sharp decline in the relative abundance of Bacteroidetes [127]. LPS and the proinflammatory cytokine-induced activation of microglia and astrocytes have been demonstrated to be exacerbated by peripheral inflammation. Walnut eating was discovered to reduce peripheral inflammation by remodeling aberrant gut microbiota, which may contribute to CNS inflammation prevention through the maintenance of a healthy gut–brain axis [174].

Interestingly, pumpkin polysaccharide altered gut microbiota composition while selectively enriching key Bacteroidetes, Prevotella, Deltaproteobacteria, Oscillospira, Veillonellaceae, Phascolarctobacterium, Sutterella, and Bilophila species, some of which are known to be reduced in ASD individuals [99,175,176,177,178]. Tamana et al. [177] reported that late-infancy gut microbiota dominated by Bacteroides is related to improved neurodevelopment, most notably in males but not in females.

Although there is significant evidence in the present research that dietary adjustments may offer treatment methods to address gut barrier dysfunction, the link between food, gut function, and disease management is still being studied and investigated. However, “leaky gut syndrome” is not totally proven, and there are numerous disagreements. Fortunately, both walnuts and pumpkin were among the fiber-rich recommended foods to treat gut leakiness [178]. Figure 7 shows the effects of walnuts and pumpkin on gut microbiota.

## 5. Conclusions

This case study does not support broad food treatments as a treatment for ASD, but it does imply that specialized dietary interventions over time may play a role in the management of certain ASD symptoms, functions, and clinical domains. The pumpkin/walnut healthy diet improved nutritional status, presumably increasing the brain’s ability to function and learn by reducing oxidative stress, neuroinflammation, glutamate excitotoxicity, mitochondrial dysfunction, and altered gut microbiota, all of which are etiological mechanisms behind the clinical presentation of ASD.

## Figures and Tables

**Figure 1 nutrients-15-04564-f001:**
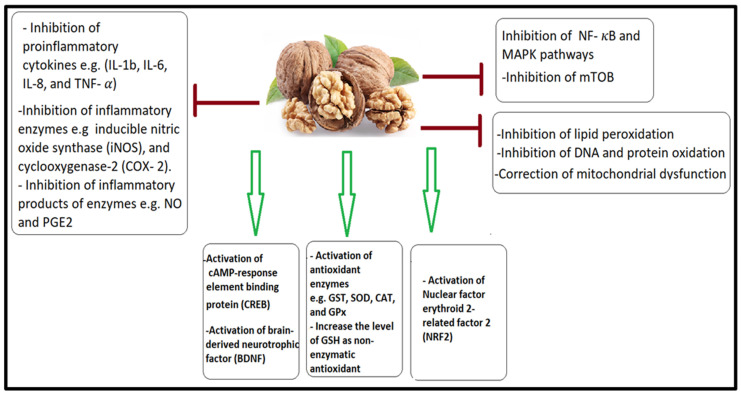
The antioxidant and anti-inflammatory properties of walnut peptides. By triggering the CREB, BDNF, and antioxidant Nrf2 signaling pathways as well as significantly enhancing the synthesis of antioxidant enzymes, walnut peptides shield cells from oxidation. Numerous brief walnut peptides stopped the death process and consequent mitochondrial dysfunction. Lipid peroxidation, pathways, and levels of inflammatory cytokines were all decreased or blocked after walnut peptide therapy.

**Figure 2 nutrients-15-04564-f002:**
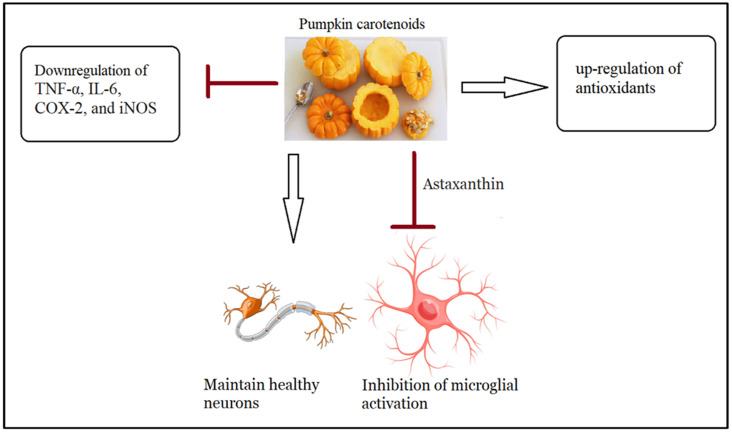
Recorded anti-inflammatory effects of pumpkin pulp.

**Figure 3 nutrients-15-04564-f003:**
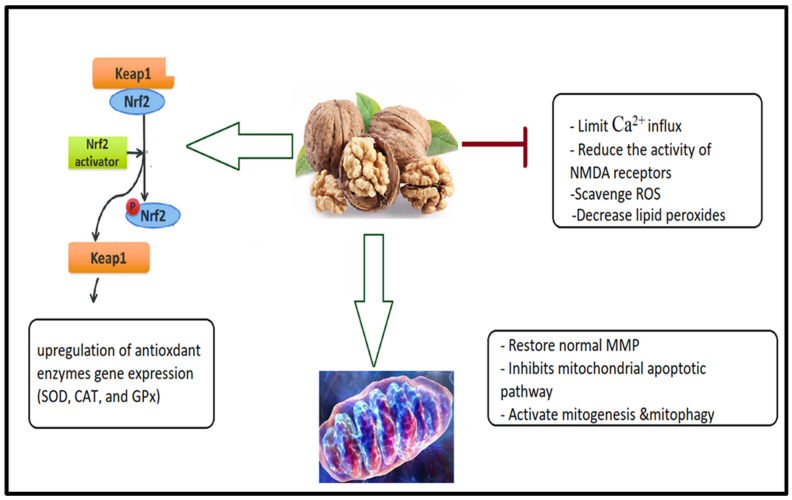
Suggested anti-excitotoxic effects of walnut peptides as treatment intervention in ASD.

**Figure 4 nutrients-15-04564-f004:**
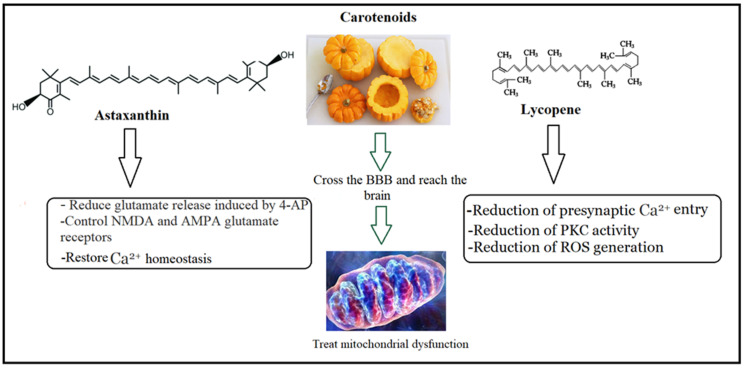
Anti-excitotoxic effects of pumpkin flesh as treatment intervention in ASD.

**Figure 5 nutrients-15-04564-f005:**
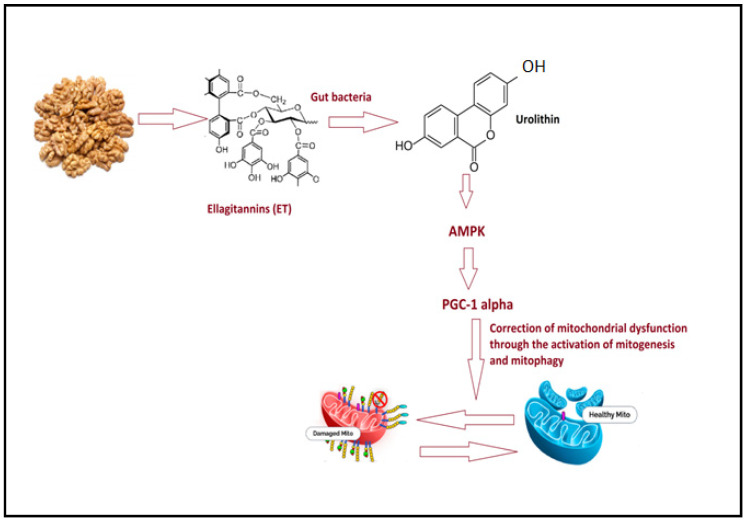
An illustration of the method by which walnut eating can treat mitochondrial dysfunction as a neurophenotype in ASD.

**Figure 6 nutrients-15-04564-f006:**
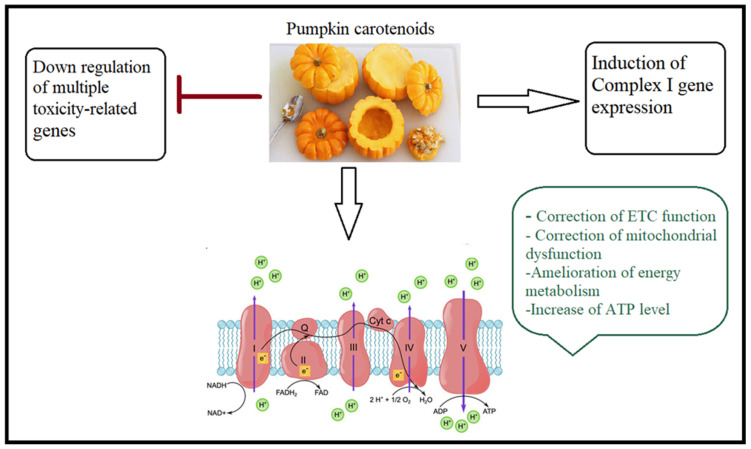
An illustration of the method by which pumpkin eating can treat mitochondrial dysfunction as a neurophenotype in ASD.

**Figure 7 nutrients-15-04564-f007:**
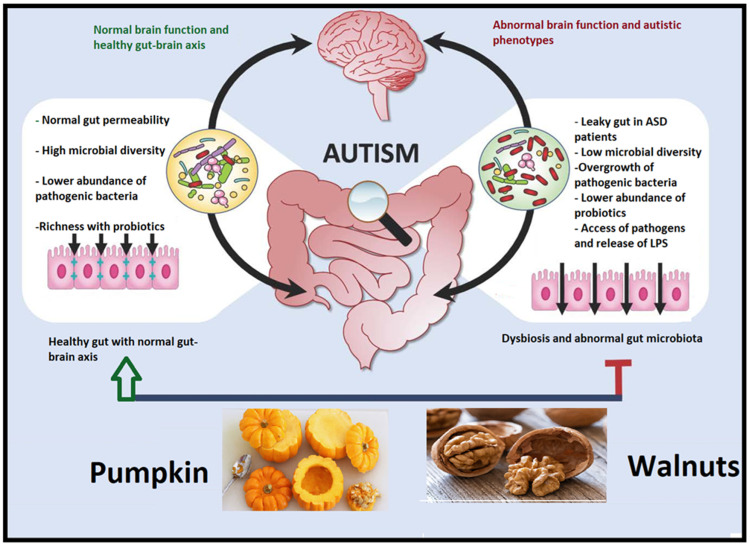
Consuming walnuts and pumpkin may reduce CNS inflammation by restructuring pathogenic gut flora and could be used to treat gut leakiness.

**Table 1 nutrients-15-04564-t001:** Improvements in CARS score during the course of the dietary intervention.

Age	CARS
2 years and 2 months	36
4 years and 6 months	27
6 years	22
13 years (now)	17 (most recent)

## Data Availability

Not applicable.

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
