# Peer review of "Effects of Walnut and Pumpkin on Selective Neurophenotypes of Autism Spectrum Disorders: A Case Study"

_nutrients, 2023, doi:10.3390/nu15214564_

Round 1
Reviewer 1 Report
Comments and Suggestions for Authors
The prescription of tailored diet plans and the use of food supplements in individuals with pervasive disorders is a topic of great scientific interest.
In their interesting review, the authors highlight how regular intake of walnuts and pumpkin contributes to reducing oxidative stress, neuroinflammation, glutamate excitotoxicity, mitochondrial dysfunction and alteration of the gut microbiota, improving the clinical presentation of the disease.
They provide an accurate description of the physiological mechanisms responsible for improving the central nervous system in general and brain function in particular.
However, I would like to suggest that the authors should better detail the patient's medical history, the starting physiological parameters and how these changed over time. In addition, they should describe in more detail the diet plan used, the daily intake (particularly of walnuts and pumpkin) and whether these changed over time.
Author Response
The prescription of tailored diet plans and the use of food supplements in individuals with pervasive disorders is a topic of great scientific interest.
In their interesting review, the authors highlight how regular intake of walnuts and pumpkin contributes to reducing oxidative stress, neuroinflammation, glutamate excitotoxicity, mitochondrial dysfunction and alteration of the gut microbiota, improving the clinical presentation of the disease.
They provide an accurate description of the physiological mechanisms responsible for improving the central nervous system in general and brain function in particular.
However, I would like to suggest that the authors should better detail the patient's medical history, the starting physiological parameters and how these changed over time. In addition, they should describe in more detail the diet plan used, the daily intake (particularly of walnuts and pumpkin) and whether these changed over time.
- Done and more detailed description of the patients’ medical history and how these changed over time was summarized in Table 1. The dietary plan was a little bit rewritten in more details through the personal communication with X-child’s mother.
Reviewer 2 Report
Comments and Suggestions for Authors
This manuscript by El-Ansary and Al-Ayadhi describes a case study of a boy with moderate autism who showed improvements in autistic symptoms upon supplementation of a diet with walnuts and pumpkin for 3 years. The authors have discussed in-depth various mechanisms such as oxidative stress, inflammation, mitochondrial dysfunction, glutamate excitotoxicity and altered gut microbiota involved in autism, that could be improved with walnuts/pumpkin diet.
1. Overall, this is well written review. However, it is based on only one case study with one boy. The authors need to discuss the limitations of the findings and avoid over-exaggerating that such a diet can be used to treat or cure autism. Dietary changes may improve the symptoms, but autism cannot be cured with any diet. Authors need to be careful in discussing the findings particularly in reference to various mechanisms. They can say that these findings are encouraging, and more case studies are needed.
2 2. Abstract and Conclusion need to be revised and more specific to the text discussed in this review article.
3. All the abbreviations when used for the first time need to be spelled out.
4 4. The manuscript needs editing. There are many incomplete sentences.
Abstract (Page 1):
1. line 10, pl. replace “used to treat” with “given to”
2. Lines 11-12, delete There is no known---- economic burden.
Page 2
Line 62, no hyphen between pep and tide
Line 65, no hyphen between transcyto and sis
Line 76, delete “and a folk remedy”.
line 108, replace “treatments” with “dietary supplements.”
Line 164, “cure story” is over-exaggeration.
Line 247, delete development of pathological disorders.
Line 264, replace ASD patients with ASD individuals or subjects.
Line 266, it is written that ASD can be corrected. Again, it is over-exaggeration.
Line 286, more references are needed to justify the statements here.
Line 294-295, delete “with a variety of changes in the brain”.
Line 295, replace neurodegenerative conditions with neurodegenerative diseases.
Line 299, replace illnesses with conditions.
Line 337, delete vegetarian.
Line 343, ref. 109 is not appropriate here.
Lines 350-354, add References to justify this sentence.
Line 408, correct neurogenerative to neurodegenerative
Line 413, delete: the treatment of
Line 427, delete “in humans.”
Comments on the Quality of English Language
The manuscript needs editing and careful reading. There are a few incomplete sentences and errors.
Author Response
Reviewer 2:
This manuscript by El-Ansary and Al-Ayadhi describes a case study of a boy with moderate autism who showed improvements in autistic symptoms upon supplementation of a diet with walnuts and pumpkin for 3 years. The authors have discussed in-depth various mechanisms such as oxidative stress, inflammation, mitochondrial dysfunction, glutamate excitotoxicity and altered gut microbiota involved in autism, that could be improved with walnuts/pumpkin diet.
- Overall, this is well written review. However, it is based on only one case study with one boy. The authors need to discuss the limitations of the findings and avoid over-exaggerating that such a diet can be used to treat or cure autism. Dietary changes may improve the symptoms, but autism cannot be cured with any diet. Authors need to be careful in discussing the findings particularly in reference to various mechanisms. They can say that these findings are encouraging, and more case studies are needed.
- Abstract and Conclusion need to be revised and more specific to the text discussed in this review article.
- Done and you can find highlighted in yellow within the text
- All the abbreviations when used for the first time need to be spelled out.
- Done and list of abbreviations was inserted, You can find highlighted in yellow within the text
- The manuscript needs editing. There are many incomplete sentences.
Abstract (Page 1):
Line 10, pl. replace “used to treat” with “given to”
- Done
Lines 11-12, delete There is no known---- economic burden.
- Done
Page 2
Line 62, no hyphen between pep and tide
-Done
Line 65, no hyphen between transcyto and sis
- Done
Line 76, delete “and a folk remedy”.
- Done
Line 108, replace “treatments” with “dietary supplements.”
- Done
Line 164, “cure story” is over-exaggeration.
- Done
Line 247, delete development of pathological disorders.
- Done
Line 264, replace ASD patients with ASD individuals or subjects.
- Done along the manuscript
Line 266, it is written that ASD can be corrected. Again, it is over-exaggeration.
- Done
Line 286, more references are needed to justify the statements here.
- Highlighted, three relevant references are already cited
Line 294-295, delete “with a variety of changes in the brain”.
- Done
Line 295, replace neurodegenerative conditions with neurodegenerative diseases.
- Done
Line 299, replace illnesses with conditions.
- Done
Line 337, delete vegetarian.
- Done
Line 343, ref. 109 is not appropriate here.
- Reference was replaced by an appropriate reference
Lines 350-354, add References to justify this sentence.
- 3 relevant references related to ALA anti-inflammatory effects are cited
Line 408, correct neurogenerative to neurodegenerative
- Done
Line 413, delete: the treatment of
- Done
Line 427, delete “in humans.”
- Done
Reviewer 3 Report
Comments and Suggestions for Authors
This is an interesting review on the role of dietary walnuts and pumpkin in the regulation of autism (ASD) pathophysiology, being exemplified in by a single case study. It is clearly written and will be interesting to your readers.
Major
Lines 410-16: It would be interesting to add something like the following and associated references, which are given as a guide:
It should be noted that generally there is an increase in serotonin in ASD, making the reference to pumpkin provision of tryptophan slightly misleading. It would be good at this point to indicate that the problem with the tryptophan-melatonin pathway in ASD seems to be the decreased ability to utilize serotonin to initiate the melatonergic pathway. Therefore, the increase in serotonin in ASD is typically associated with a decrease in melatonin across different body cells, including pinealocytes, platelets and intestinal epithelial cells [Pagan et al., 2017]. This is highly likely to be associated with the alterations in mitochondrial function in ASD [Maes et al., 2019], given that the melatonergic pathway is present in the mitochondria of all body cells investigated to date. The attenuated availability of the melatonergic pathway in ASD seems to arise from a decrease in 14-3-3 isoforms due to differential regulation by microRNAs (including miR-451). As walnuts and pumpkin significantly regulate a number of miRNAs [Gil-Zamorano et al., 2022; Lambertini et al., 2023], some of the efficacy of walnuts my be linked to the regulation of the tryptophan-melatonin pathway.
References
Pagan C, Goubran-Botros H, Delorme R, Benabou M, Lemière N, Murray K, Amsellem F, Callebert J, Chaste P, Jamain S, Fauchereau F, Huguet G, Maronde E, Leboyer M, Launay JM, Bourgeron T. Disruption of melatonin synthesis is associated with impaired 14-3-3 and miR-451 levels in patients with autism spectrum disorders. Sci Rep. 2017 May 18;7(1):2096. doi: 10.1038/s41598-017-02152-x.
Maes M, Anderson G, Betancort Medina SR, Seo M, Ojala JO. Integrating Autism Spectrum Disorder Pathophysiology: Mitochondria, Vitamin A, CD38, Oxytocin, Serotonin and Melatonergic Alterations in the Placenta and Gut. Curr Pharm Des. 2019;25(41):4405-4420. doi: 10.2174/1381612825666191102165459.
Gil-Zamorano J, Cofán M, López de Las Hazas MC, García-Blanco T, García-Ruiz A, Doménech M, Serra-Mir M, Roth I, Valls-Pedret C, Rajaram S, Sabaté J, Ros E, Dávalos A, Sala-Vila A. Interplay of Walnut Consumption, Changes in Circulating miRNAs and Reduction in LDL-Cholesterol in Elders. Nutrients. 2022 Apr 1;14(7):1473. doi: 10.3390/nu14071473.
Lambertini E, Penolazzi L, Notarangelo MP, Fiorito S, Epifano F, Pandolfi A, Piva R. Pro‑differentiating compounds for human intervertebral disc cells are present in Violina pumpkin leaf extracts. Int J Mol Med. 2023 May;51(5):39. doi: 10.3892/ijmm.2023.5242.
vvv
Author Response
Reviewer 3:
Major
Lines 410-16: It would be interesting to add something like the following and associated references, which are given as a guide:
It should be noted that generally there is an increase in serotonin in ASD, making the reference to pumpkin provision of tryptophan slightly misleading. It would be good at this point to indicate that the problem with the tryptophan-melatonin pathway in ASD seems to be the decreased ability to utilize serotonin to initiate the melatonergic pathway. Therefore, the increase in serotonin in ASD is typically associated with a decrease in melatonin across different body cells, including pinealocytes, platelets and intestinal epithelial cells [Pagan et al., 2017]. This is highly likely to be associated with the alterations in mitochondrial function in ASD [Maes et al., 2019], given that the melatonergic pathway is present in the mitochondria of all body cells investigated to date. The attenuated availability of the melatonergic pathway in ASD seems to arise from a decrease in 14-3-3 isoforms due to differential regulation by microRNAs (including miR-451). As walnuts and pumpkin significantly regulate a number of miRNAs [Gil-Zamorano et al., 2022; Lambertini et al., 2023], some of the efficacy of walnuts my be linked to the regulation of the tryptophan-melatonin pathway.
- Thanks a lot for your appreciated and professional comment which will increase the scientific merit of our review. It was done and the four recent suggested references were cited. Again thanks